# A New Data-Preprocessing-Related Taxonomy of Sensors for IoT Applications

Paul D. Rosero-Montalvo [1,*,†], Vivian F. López-Batista [2,†] and Diego H. Peluffo-Ordóñez [3,4,†]

1 Computer Science Department, IT University of Copenhagen, 2300 Copenhagen, Denmark
2 Department of Computer Science and Automatics, University of Salamanca, 37008 Salamanca, Spain; vivian@usal.es
3 Morocco and SDAS Researh Group, Modeling, Simulation and Data Analysis (MSDA) Research Program, Mohammed VI Polytechnic University, Ben Guerir 43150, Morocco; diego.peluffo@sdas-group.com or diego.peluffo@aunar.edu.co
4 Faculty of Engineering, Corporación Universitaria Autónoma de Nariño, Pasto 520001, Colombia
* Correspondence: puldavid87@gmail.com
† These authors contributed equally to this work.

**Abstract:** IoT devices play a fundamental role in the machine learning (ML) application pipeline, as they collect rich data for model training using sensors. However, this process can be affected by uncontrollable variables that introduce errors into the data, resulting in a higher computational cost to eliminate them. Thus, selecting the most suitable algorithm for this pre-processing step on-device can reduce ML model complexity and unnecessary bandwidth usage for cloud processing. Therefore, this work presents a new sensor taxonomy with which to deploy data pre-processing on an IoT device by using a specific filter for each data type that the system handles. We define statistical and functional performance metrics to perform filter selection. Experimental results show that the Butterworth filter is a suitable solution for invariant sampling rates, while the Savi–Golay and medium filters are appropriate choices for variable sampling rates.

**Keywords:** Internet of Things; sensor; machine learning; computational intelligence; data analytics; data pre-processing

## 1. Introduction

Internet of Things (IoT) technology allows electronic devices to be deployed in indoor and outdoor environments to collect data [1]. Commonly, these IoT devices consist of a microcontroller, sensors, a battery, and wireless communicationelectronic devices to be deployed in indoor and outdoor environments to collect data. IoT devices can be installed in harsh scenarios due to their flexible development [2]. Nowadays, about 22 billion IoT devices are uploading data to the cloud. Every year, this number increases exponentially to continue collecting data through a wide variety of sensors. These data are used to train machine learning (ML) models, powerful tools that can find hidden knowledge in data that describes a phenomenon or human behavior [3]. However, constantly uploading data to the cloud causes bottlenecks in the communication channel, and in some cases, the stored data are not processed for a specific purpose [4]. Hence, cloud computing servers have to delete data periodically to avoid storage overload. Consequently, data quality is essential to reduce the complexity of the ML model, and it is necessary to send only relevant data to be processed. Therefore, after the data gathering process, a data pre-processing step is required to eliminate errors, which means both stages are part of the ML pipeline [5]. There are several repositories in different areas where researchers and developers can obtain databases to deploy and test ML models. They assume that the data are cleaned before being put into the repository. Nevertheless, this is not the case for IoT environments,

where sensors gather data in situ because they describe specific parameters such as environmental conditions, gas concentration, and location. In conclusion, data gathering and pre-processing are obligatory stages of building an ML application in IoT environments.

The data collection stage in IoT environments needs to handle uncontrolled conditions such as environmental changes, construction failures of microcontrollers and sensors that cause poor calibration, and vibrations in their working environment, among others [6]. Therefore, model inference can reduce performance, resulting in the use of complex models when describing a phenomenon or human behavior [7]. The pre-processing stage produces reliable, accurate, repeatable, and error-free data [8]. Thus, the electrical signal obtained by the sensors should be acquired with an adequate sampling rating and proper tuning of analog-to-digital converters [9]. On the software side, digital filters are applied when data have been stored on servers. However, sensors have different data collecting procedures, such as digital-analog converters, communication ports, and pulse trains [10]. Therefore, the cloud can not apply a standard filtering process to all the features stored. Additionally, this data flow over communication channels increases security concerns and decreases user confidence in the system. Therefore, new computational paradigms propose decentralized computing where some ML stages run closer to the user, which means performing data-preprocessing locally [11]. It is worth pointing out that these algorithms can be deployed on IoT devices due to the microcontrollers' increasing computing capacity, which will not affect battery consumption [12]. Additionally, sensors vendors are working to give the IoT developer robust libraries to improve sensor management [13]. However, sensor data need to be pre-processed before sending it to the cloud [14].

Data filtering removes noise by comparing each signal component to the rest and eliminating the unusual ones. The most relevant criteria and their principal algorithms are infinite impulse response (IIR) with the approximations Butterworth, Bessel, and Chebyshev; finite impulse response (FIR) with the windows Hamming, Tayler, Barlett, and Blackman; and smoothing filters with the algorithms: mean, average, Gaussian, and Savi–Golay [14]. For more information about digital filter design, we suggest following these works [15,16]. These filter criteria depend on the sampling rate at which the IoT device is configured, the collection procedure of each sensor, and the application. However, previous sensor taxonomies focus on hardware characteristics without considering their primary purpose of collecting data. In addition, data filtering criteria are applied for each IoT development, which consumes additional time for IoT researchers and developers.

It is necessary to define a new sensor taxonomy related to the data collection and pre-processing processes that fits the filtering criteria to be part of the whole ML pipeline [17]. Therefore, this work introduces a new sensor taxonomy oriented to pre-processing data on-device according to the type of sensor used in the IoT application. Consequently, we need to define how IoT devices collect data through sensors to determine the suitable filter for each case. Our summarized contributions are:

- We define a new sensor taxonomy related to data gathering and data pre-processing on-device.
- We determined that the main sensor characteristic for classification is sampling rate.
- We introduce a data filtering scheme using the most representative algorithms/models of infinite impulse response (IIR), finite impulse response (FIR), and smoothing filters by setting specific sampling rates for each sensor type.
- We compare data filtering criteria to select the suitable ones for the proposed taxonomy of sensors and ensure its usefulness in computationally constrained IoT environments.
- We performed tests on sensor data with statistical and functional metrics.

The main result of this work is defining the Butterworth filter as a suitable criterion for analog sensors with invariant sampling rates. Meanwhile, Savi–Golay fits analog sensors with varying sampling frequencies. The average filter is adequate with this signal in digital pulse train sensors. Savi–Golay and medium filters remove noise and preserve the main signal characteristics regarding communication protocol sensors.

The rest of the manuscript is structured as follows: Section 2 shows related works and signal filtering background. Then, Section 3 introduces the proposed sensor taxonomy. Next, the methodology is shown in Section 4. Results are presented in Section 5 with the statistical and sensor functionality metrics to define the filter algorithm we chose. Finally, Section 6 shows conclusions and future work.

## 2. Background

In this section, we present a summary of previous sensor taxonomy and data filtering works.

### 2.1. Early Studies Sensors

The increasing use of electronic devices in the industry has opened up opportunities to develop different types of sensors. Indeed, new technology trends such as the Internet of Things (IoT) allowed expanding the research areas where sensors are used. Therefore, new ways to describe/classify them are relevant, since they play a significant role in the data-gathering stage of the entire machine learning application pipeline. Thus, in the early stages of sensor development, works such as MacRuairi et al. [18] presented sensor requirements taxonomies to match specific sensors with real scenarios. Then, Fowler et al. [19] presented a survey related to the materials that sensors are made from. Following this classification scheme, works such as Tuukkanen et al. [20], Noel et al. [21], Cornacchia et al. [22], and Khanh et al. [23] presented sensors surveys for specific areas, such as piezoelectric sensors, health monitoring, wearable sensors, and intelligent agriculture, respectively. In recent years, Abdel Azeem et al. [17] have shown the fundamentals, challenges, opportunities, and taxonomy of sensors in IoT environments describing the needs and usages of each one. They also presented a wide array of previously proposed solutions, comparing them to each other and providing brief descriptions of the issues addressed by each category of that taxonomy. Finally, works such as Latifi et al. [24] and Anajeba et al. [25] presented early intuitions about improving the security of the communication channel in IoT environments.

In the ML application pipeline, Morrison et al. [26] present an innovative survey in sensor data collection and analytical systems. Additionally, Infanteena et al. [27] showed a survey on compressive data collection techniques for IoT devices and analyzed their features. Finally, in this research area, Tiboni et al. [28] described sensors and actuators in exoskeletons using the machine learning pipeline.

### 2.2. Data Pre-Processing

The most relevant works in this field started with Zhang et al. [14] presenting a relevant work about a data $H_\infty$ filtering approach for wireless sensor networks (WSNs) in nonuniform sampling periods with optimization techniques. Then, Deepshukha et al. [29] designed a low-power digital FIR filter on FPGA for noise reduction in a WSN. Later, Bose et al. [2] presented an analysis of contemporary lossy compression algorithms using the signal characteristics of sensor data. At the same time, Safaei et al. [30] showed a novel approach to integrating time-series analysis, entropy, and random forest-based classification. For their part, Kowalski et al. [31] presented a review and comparison of smoothing algorithms for one-dimensional data noise reduction in specific sensors and environments. Timo et al. [12] presented outlier detection from non-smooth sensor data, as they worked in spatial discontinuities in the data, such as those arising from shadows in photovoltaic (PV) systems. Saad et al. [32] analyzed how quantization affects distributed graph filtering over both time-invariant and time-varying graphs. We bring insights into the quantization effects of the two most common graph filters: the finite impulse response (FIR) and auto-regressive moving average (ARMA) graph filters. In addition, we have proposed robust filter design strategies that minimize the quantization noise for time-invariant and time-varying networks.

Several works have delved into data pre-processing in IoT devices, but most introduced approaches for specific scenarios without a rationale for the selected filter criterion.

On the one hand, outlier detection is a complex task for determining if external causes have corrupted the data. Therefore, as mentioned in [33], the filtering process must be carried on before the outlier detection stage. On the other hand, filtering can avoid physical constraints by giving a clean dataset to implement different stages. Finally, the literature review allowed us to observe open challenges in data filtering, such as the lack of a sensor taxonomy related to the data acquisition process and the establishment of adequate sampling rates for each type of sensor.

## 3. Proposed Sensor Taxonomy

We propose classifying sensors into three groups considering the sampling rate and how sensors send information to the microcontroller. Figure 1 illustrates this taxonomy.

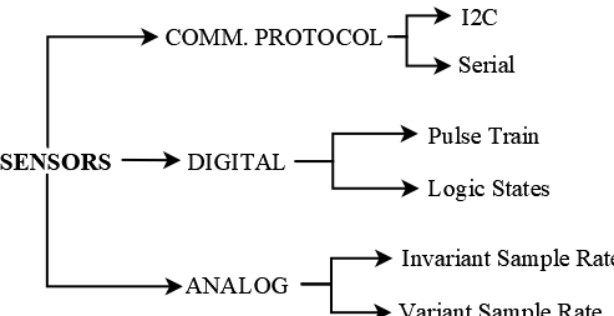

**Figure 1.** Proposed taxonomy of IoT sensors considering data processing characteristics.

### 3.1. Analog Sensors

These sensors mostly have passive elements and operational amplifiers for the hardware conditioning of the electrical signal and deliver it analogously to the microprocessor to convert it to digital form (analog–digital conversion) [34]. The ability to recreate the original signal is related to the resolution of the ADC, which is the number of bits that the microprocessor has for this process. Therefore, the sampling rate is the most relevant characteristic of the filter implementation criteria. Hence, we divide them into two categories:

- Invariant sampling rate: These sensors are developed for collecting signals continuously to detect changes in a main characteristic. For example, the processing of human electrical activity through electromyography (muscle), electrocardiogram (heart), electroencephalogram (EEG), or galvanic skin response (hands).
- Variant sampling rate: These sensors run a couple of times a day due to their applications. They do not have a specific sampling frequency because the system focuses on taking the same number of samples each time it is activated [6].

### 3.2. Digital Sensors

These sensors each contain a tiny microcontroller to perform the ADC process by themselves and send the data to the main microcontroller in two ways:

- Pulse train sensors: variate their pulse train frequency when the transducer detects that a physical magnitude such as temperature, humidity, or distance is changing. Therefore, capacitors are often used in this type of sensor.
- Logic states sensors: use only two logical values, 3.3 vs. or 5 vs., when detecting a physical magnitude, no matter their variations, and 0v when the sensor cannot catch the magnitude. Thus, for example, the human presence sensor can not give us more information about the phenomenon, just its presence.

### 3.3. Sensors by Communication Protocol

They are the most complex sensors because they have a microcontroller inside whose main objective is to obtain the best signal of the physical magnitude. These sensors also implement a communication protocol to connect sensors in series. Therefore, only a few

pin connections are necessary to handle many sensors. Furthermore, these communication protocols define a master device (microcontroller) to coordinate the slave devices' (sensors) communication. Nowadays, sensor vendors, such as SparkFun, perform new socket connections to develop the electronic systems quickly.

- Serial communication: A sensor uses one pin to transmit messages and another pin to receive them. This protocol extensively adds wireless protocols to the IoT device, such as Bluetooth.
- $I_2C$: They have a new socket connection called Qwiic (Connect System uses 4-pin JST connectors to quickly interface development boards with sensors). This standard also allows connecting 127 sensors using just two pins. One is the clock rate, and the other is the transmitter line.

## 4. Methodology

The proposed methodology determines the sensors used and the data sampling required to implement filters. First, it is necessary to mention that the FIR and IIR filters are implemented only in the sensors with invariant sampling rates and the signal smoothing technique on the rest. However, the metrics used for both criteria are: signal-to-noise ratio (SNR), mean squared error (MSE), mean absolute error (MAE), root-mean-square error (RMSE), and R2 score. Figure 2 shows the mentioned process.

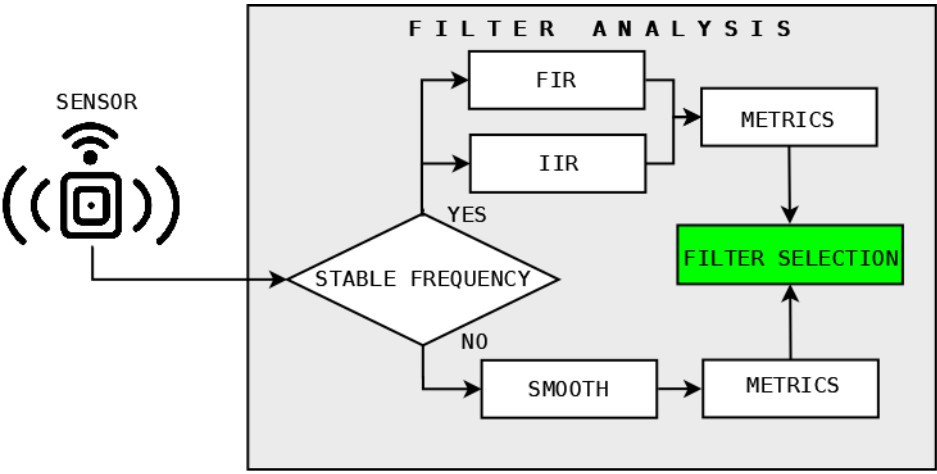

**Figure 2.** Sensor data and pre-preprocessing analysis.

### 4.1. Sensors' Characteristics

The most commonly used sensors were identified from the reviewed related works. As a result, the relevant research areas are smart farming, cities' environmental conditions analysis, and human illness. Therefore, sensors chosen regarding the proposed taxonomy were ECG Pulse Sensor (Bio-signal), Force Sensitive Resistor (FSR) (specific propose), Flex Sensor (Specific propose), Humidity and Temperature Sensor DHT-22 (pulse train), Gas Sensor MQ-135 (pulse train), and $CO_2$ sensor-SCD30 ($I_2C$/serial), UV sensor-VEML6075 ($I_2C$/serial). These sensors are from the same sensor vendor company SparkFun. We avoided using logical state sensors because they would not allow us to have data filtering criteria with only two values. Moreover, the sensors' communication protocol offers us the same ability to use $I_2C$ and serial protocol. Table 1 shows the principal characteristics of each sensor used.

**Table 1.** Most commonly used sensors in IoT devices regarding the proposed taxonomy.

| Sensor Type | Sensor | Characteristics |
|---|---|---|
| Bio-Signals | ECG (pulse sensor) | Detects changes in the volume of a blood vessel that occur when the heart pumps blood. To do so, they emit infrared, red or green light (550 nm) towards the body and measure the amount of reflected light with a photodiode or phototransistor. It has an operating voltage between 3.3 and 5 volts with a power consumption of 4 mA. |
| Specific Propose | Flexometer | Produces a variable resistance according to the degree to which it is bent. In this sense, the sensor converts the bending into different values of electrical resistance. |
| | Force | The force-sensing resistance sensor (also called FSR) varies its internal resistance when pressure is applied to its sensing area. As of this effect, the output voltage changes as well. Thus, the higher the pressure, the higher the output voltage. |
| Pulse train | Humidity and Temperature (DTH11) | This sensor sends a calibrated digital signal containing an 8-bit microcontroller. In addition, it contains two resistive sensors (NTC and humidity). It uses one-wire communication (pulse train). |
| | gas NOx (MQ135) | This air quality sensor detects gas concentration in various percentages. The output signal presents TTL voltage levels to be processed by a microcontroller. |
| Cx I2C | $CO_2$ (SCD 30) | This is a high quality non-dispersive infrared (NDIR) based $CO_2$ sensor capable of detecting from 400 to 10,000 ppm with an accuracy of $\pm$ (30 ppm + 3%). |
| | UV (VEML) | This sensor implements a simple photodiode to measure UVA (320–400 nm) and UVB (280–320 nm) radiation levels. With this data, it can read the intensity of these types of light in irradiance and, from there, calculate the UV index. |

*4.2. Data Samples Acquisition*

First, we started with the ECG Pulse Sensor of the invariant sample rate sensors. The sample rate was 1 kHz (Nyquist theorem) because the signal has main components until 100 Hz. Therefore, 1400 samples were obtained in 10 controlled experiments. Second, the variable sample rate sensors were exposed to their physical magnitude for 10 s, and then they returned to their initial condition (flexometer and force sensors). Consequently, this process was carried out ten times to store 1000 samples with a 100 Hz sample rate. A similar procedure was carried out with pulse train sensors, such as DTH11 and MQ135. Finally, communication protocol sensors (SCD30 and VMLE) were tested in 10 controlled experiments. As a result, we stored 500 samples with a 50 Hz sample rate because their response times are higher than those of the other sensors.

**5. Results**

The sensors were tested with statistical metrics according to the experiments performed for each one. Then, they were evaluated with functional metrics such as accuracy, reproducibility, repeatability, and stability. These metrics represent the sensors working in real conditions. Thus, for a better understanding of each metric result, four evaluation levels were established for the sensors: (i) excellent, (ii) good, (iii), normal, and (iv) poor.

Finally, Figure 3 shows all the sensors used in this research and their connections with a sample board, such as Arduino.

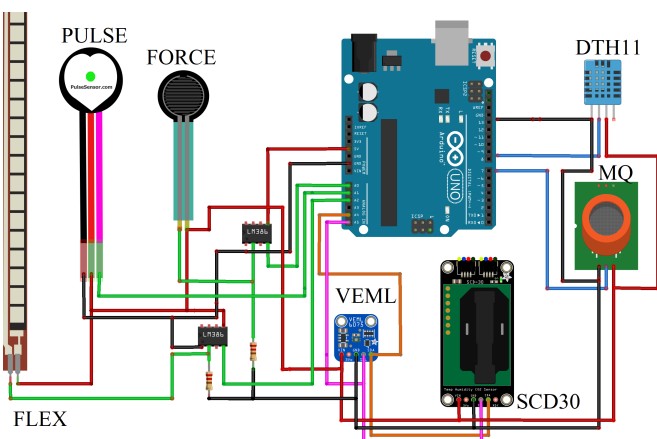

**Figure 3.** Sensors used in this work according to the new proposed taxonomy. (—) Analog connection, (—) Digital connection, (—) Prot. Communication connection (SDA), ( —) Prot. Communication connection (SCL), (—) VCC, (—) GND.

### 5.1. Invariant Sampling Rate (ISR)

The signal needs to be converted to the frequency domain to detect the principal components. Therefore, a fast Fourier transform was implemented to define that the EMG components were between 5 and 40 Hz, which are presented in Figure 4. Then, IIR filters were the first approach with **Chebyshev**, **Butterworth**, and **Bessel** approximations with 3, 5, and 7 orders of band-pass filter design. We noticed that the filters in order 5 fit better than the rest. Table 2 summarizes the results of the statistical metrics mentioned before. The **Butterworth** filter demonstrated superior SNR, MAE, and R2 metrics. Additionally, it is visible that **Butterworth** reduced the noise with few signal alterations. The second approach was FIR filters. They focus on a time-domain analysis through windows. Reference [35] defines that using 10% as a window size of sample rate is recommended. Thus, we defined window sizes of 150, 250, and 300 components to compare with the ECG signal. Table 3 summarizes that windows size equal to 150 components produced a better SNR when **Nutall window** was applied. However, the differences between the windows were minimal when we tried to improve the signal. As a result, FIR filters are a better option than IIR. Finally, Figure 4 shows the components in the frequency domain and the graphical results of FIR and IIR filters.

**Table 2.** EMG signal statistical analysis and IIR filters.

| Approximation | SNR (dB) | MSE | MAE | RMSE | R2 |
|---|---|---|---|---|---|
| Butterworth | 4.44 | 0.13 | 0.31 | 0.36 | −6.83 |
| Bessel | 4.20 | 0.20 | 0.38 | 0.44 | −10.66 |
| Chebyshev | 4.12 | 0.12 | 0.30 | 0.34 | −6.26 |

**Table 3.** EMG signal statistical analysis and FIR filters.

| Window | SNR (dB) | MSE | MAE | RMSE | R2 |
|---|---|---|---|---|---|
| Nutall | 4.48 | 0.04 | 0.18 | 0.20 | −1.55 |
| Hamming | 3.77 | 0.13 | 0.33 | 0.36 | −7.15 |
| Taylor | 4.21 | 0.80 | 0.81 | 0.9 | −8.43 |
| Blackman | 4.09 | 0.06 | 0.22 | 0.25 | −3.0 |

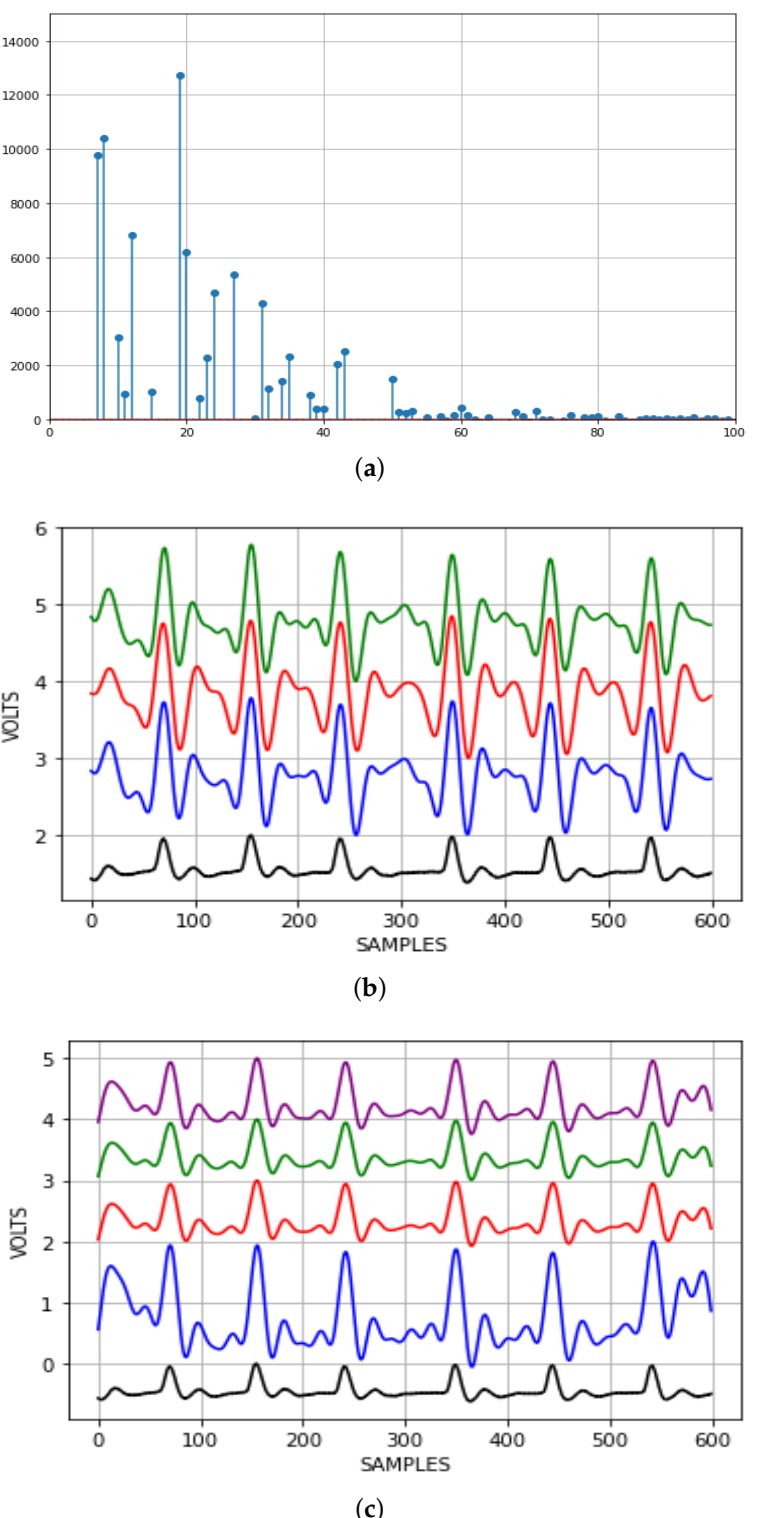

**Figure 4.** EMG signal analysis. (**a**) EMG signal in the frequency domain. (**b**) IIR filters: (——) Butter-woth, (——) Chebyshev, (——) Bessel, (——) original samples. (**c**) FIR filters: (——) Hamming, (——) Nutall, (——) Taylor, (——) Blackman, (——) original samples.

### 5.2. Variable Sample Rate (VSR)

For experimental purposes, the Force Sensitive Resistor sensor was tested with 40 lbs of pressure, and the Flex sensor bent it 45 degrees. Both were tested with the sample rate mentioned above (100 Hz). Additionally, their datasheet recommends using an analog

amplifier in follow-up configuration to avoid DC voltage. Therefore, we applied smoothing filters. The average filter has a better SNR metric; however, the R2 score indicates that this filter affects the original signal. Additionally, the Gaussian filter tends to round off the maximum values obtained and modifies the output signal due to `sigma` parameter (Gaussian bell size). The **Savi–Golay** filter eliminates noise in VSR signals: see the strong results in R2 score and SNR metrics (Table 4). Figure 5 shows the graphical results of each smoothing filter.

**Table 4.** Statistical analysis of sensors with various sampling rates.

| Sensor | Average k = 20 | Medium k = 20 | Gaussian Sigma = 7 | Savi–Golay k = 9, Poly = 4 | Statistical Metrics |
|---|---|---|---|---|---|
| | 9.07 | 8.28 | 8.97 | 7.90 | MSE |
| | 1.49 | 1.60 | 0.65 | 1.27 | MAE |
| Flex sensor | 1.91 | 1.96 | 0.98 | 2.81 | RMSE |
| | 0.642 | 0.56 | 0.99 | 0.99 | R2 score |
| | 2.65 | 2.16 | 2.47 | 2.49 | SNR |
| | 195.39 | 198.23 | 205.31 | 158.2 | MSE |
| | 5.25 | 5.29 | 3.20 | 4.96 | MAE |
| Force sensor | 18.85 | 15.78 | 14.32 | 15.67 | RMSE |
| | 0.75 | 0.65 | 0.99 | 0.99 | R2 score |
| | 2.91 | 2.65 | 2.86 | 2.87 | SNR |

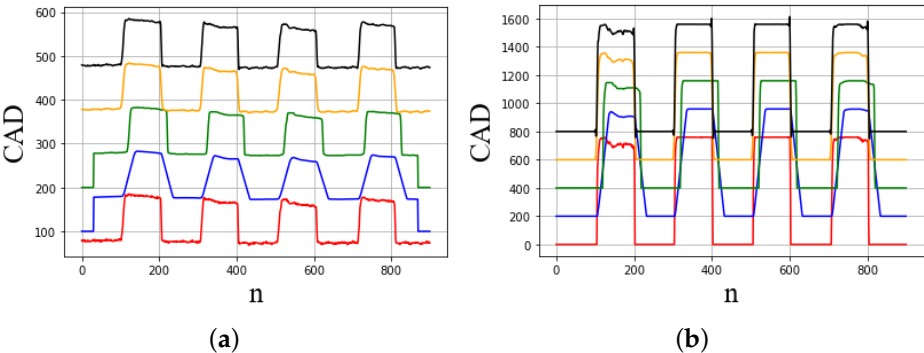

**Figure 5.** Smoothing graphical analysis in the proposed sensor taxonomy. (—) Original samples. (—) Average filter. (—) Medium filter. (—) Gaussian filter. (—) Savi–Golay filter. (**a**) FLEX sensor. (**b**) FORCE sensor.

*Sensor performance metrics:* These sensors have a variable resistor as their main component. Therefore, they are stable in operation, and similar data can be obtained in each data gathering process. However, their wear and tear is very high, subject to human activity. For this reason, they are dependent on their location and use, and their reproducibility tends to decrease over time.

### 5.3. Digital Pulse-Train

The data collection process was based on having a closed box with an incandescent bulb, a fan, and extra space for sensors. First, we used the `DTH11` to get measurements when the temperature inside the box increased due to the bulb and then decreased when the fan was powered. Then, for the gas sensor MQ135, we used a gas emitter (lighter) instead of the bulb and a fan, and a sensor inside to change the gas concentration inside quickly. These experiments demonstrated that Savi–Golay and **average** filters fit with these kinds

of signals and have better SNR metrics. Consequently, we noticed that the average filter reduces the dc voltage (peaks), producing good R2 score, MAE, and MSE results (Table 5). Moreover, Figure 6 represents the smoothing signal applied in pulse train sensors, from which we can notice that medium and Savi–Golay filters do not modify the electric signal.

*Sensor performance metrics:* These sensors have standard accuracy and stability due to their calibrated modes. However, they have restrictions on repeatability and reproducibility metrics because they sense physical magnitudes that do not vary in short periods, such as temperature and humidity, among others.

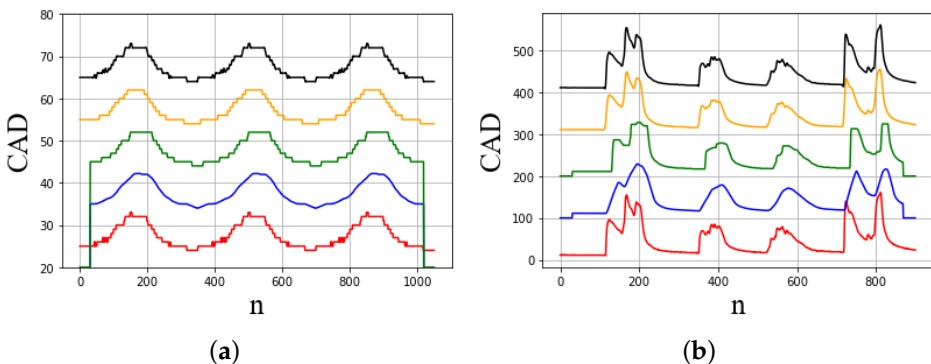

(a)                                                         (b)

**Figure 6.** Smoothing graphical analysis in the proposed sensor taxonomy. (—) Original samples (—) Average filter. (—) Medium filter. (—) Gaussian filter. (—) Savi–Golay filter. (**a**) DHT-22 sensor. (**b**) MQ-135 sensor.

**Table 5.** Digital pulse-train sensors' statistical analysis.

| Sensor | Average k = 30 | Medium k = 30 | Gaussian Sigma = 7 | Savi–Golay k = 9, Poly = 4 | Statistical Metrics |
|---|---|---|---|---|---|
| DHT-11 | 6.40 | 6.51 | 0.2 | 4.29 | MSE |
| | 2.15 | 2.14 | 0.07 | 0.29 | MAE |
| | 1.03 | 1.04 | 0.15 | 0.54 | RMSE |
| | 0.75 | 0.77 | 0.99 | 0.96 | R2 score |
| | 9.72 | 9.60 | 9.61 | 9.69 | SNR |
| MQ-135 | 13.55 | 11.79 | 10.48 | 13.73 | MSE |
| | 1.35 | 2.29 | 0.29 | 1.62 | MAE |
| | 3.77 | 3.49 | 0.69 | 3.70 | RMSE |
| | 0.51 | 0.35 | 0.99 | 0.98 | R2 score |
| | 1.36 | 1.27 | 1.26 | 1.28 | SNR |

*5.4. I2C Communication Protocol*

These sensors were exposed to their corresponding physical features (UV rays and $CO_2$ gas). Gausian and Savi–Golay filters removed the noise better than the other algorithms. However, the Gausian modifies the signal output significantly. Additionally, the average does not fit with these types of electrical signals due to the sizes of their windows affecting the signal with few samples of data. Therefore, **medium** and **Savi–Golay** can be applied to these sensors. Table 6 represents the statistical analysis, and Figure 7 shows the graphical results.

*Sensor performance metrics:* They have poor repeatability and reproducibility because UV rays do not have considerable variability during the day. Moreover, $CO_2$ can increase exponentially in fires, smoking zones, etc., but it needs a few hours to normalize. As a

result, the sensor has restrictions concerning returning to the initial state. Figure 7 shows the smoothing graphical results of both sensors.

**Table 6.** Communication protocol sensors' statistical analysis.

| Sensor | Average k = 30 | Medium k = 20 | Gaussian Sigma = 7 | Savi–Golay k = 9, Poly = 4 | Statistical Metrics |
|---|---|---|---|---|---|
| | 540.16 | 650.66 | 435.10 | 475.0 | MSE |
| | 47.89 | 69.5 | 62.14 | 105.78 | MAE |
| SCD30 | 178.05 | 111.02 | 124.23 | 182.96 | RMSE |
| | 0.55 | 0.77 | 0.94 | 0.86 | R2 score |
| | 1.51 | 1.47 | 2.01 | 2.37 | SNR |
| | 2.51 | 3.44 | 2.14 | 2.43 | MSE |
| | 0.97 | 0.98 | 0.39 | 0.9 | MAE |
| VEML6075 | 1.58 | 1.85 | 1.2 | 0.20 | RMSE |
| | 0.42 | 0.22 | 0.10 | 0.99 | R2 score |
| | 1.0 | 0.88 | 0.89 | 0.92 | SNR |

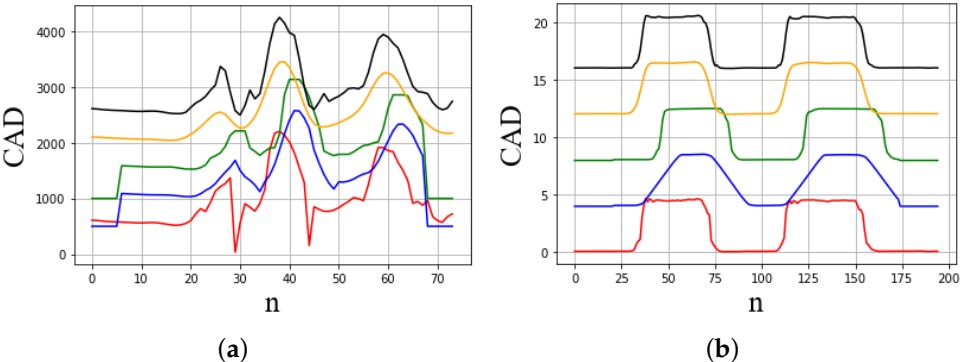

(**a**)  (**b**)

**Figure 7.** Smoothing graphical analysis in the proposed sensor taxonomy. (—) Original samples. (—) Average filter. (—) Medium filter. (—) Gaussian filter. (—) Savi–Golay filter. (**a**) SCD30 sensor. (**b**) VEML6075.

*5.5. Real Tests*

Sensors were evaluated under natural conditions to test each filter selected. In addition, we compare the voltage obtained through sensors using a multimeter KEYSIGHT DIGITAL MULTIMETER U1282A, which has a 0.025% voltage accuracy. Therefore, for a better understanding of each metric's result, four levels of evaluation were established for the sensors: (i) excellent, (ii) good, (iii), normal, and (iv) poor. Table 7 shows the results obtained.

Finally, we obtained the system response time for each sensor with the filter deployed on the device. For example, the Butterworth filter takes 2.5 ms to process an array with 300 samples, the Savi–Golay takes 1.2 ms to process the same number of samples, and the medium filter takes 0.68 ms. Therefore, this pre-processing technique is a suitable solution to run in real-time scenarios when the IoT system can define threads for each procedure to reduce the time response of each task. Additionally, filters have a small footprint in memory, leaving enough space to run the IoT application.

**Table 7.** Sensor performance metrics.

| Performance Metrics | Sensor Taxonomy | | | |
| --- | --- | --- | --- | --- |
| | Analog Sensors | | Pulse | Comm. |
| | ISR | VSR | train | Protocol |
| Accuracy | Good | Good | Normal | Excellent |
| Reproducibility | Good | Poor | Poor | Excellent |
| Repeatability | Good | Normal | Excellent | Poor |
| Stability | Normal | Poor | Good | Normal |
| Noise | Poor | Normal | Good | Good |

## 6. Conclusions and Future Works

This work introduced a new taxonomy of sensors focused on data pre-processing on-device to upload reliable data to the cloud. Furthermore, filter implementation criteria were established to prevent erroneous data from being part of the ML model. We now present the conclusions of this work:

- This taxonomy of sensors is appropriate for the new trend of executing some ML stages on-device. Therefore, this work prevents data that do not describe the phenomenon being studied from being part of the ML model. Thus, the sampling frequency used in the sensors is a fundamental part of implementing filters.
- The proposed methodology demonstrates which filter is adequate and does not deform the original signal.
- Performance metrics in real environments define the ability to reduce noise and provide new trends to improve this process for coming sensors.
- We declare the Butterworth filter suitable for analog sensors with invariant sampling rates. Savi–Golay fits analog sensors with variant sampling rates. The average filter is adequate for digital pulse train sensors. Regarding communication protocol sensors, Savi–Golay and medium filters remove noise and provide improved signal for the proposed data gathering.

Finally, we understand that the next step is to detect anomalies in sensor data due to manipulation or sensor failure.

**Author Contributions:** P.D.R.-M.: conceptualization, methodology, software, formal analysis, investigation, writing—original draft preparation, visualization, and resources; V.F.L.-B.: investigation, supervision, and project administration; D.H.P.-O: formal analysis, writing—review, visualization, project administration, and funding. All authors read and agreed to the published version of the manuscript.

**Funding:** This research was funded by Novo Nordisk Fonden, grant number NNF20OC0064411, with the project Privacy through Co-Design for Real-World Data Analytics in the cloud.

**Institutional Review Board Statement:** Not applicable.

**Informed Consent Statement:** Not applicable.

**Data Availability Statement:** Not applicable.

**Acknowledgments:** The authors are grateful for the support given by the SDAS Research Group (https://sdas-group.com/, accessed on 10 April 2022).

**Conflicts of Interest:** The authors declare no conflict of interest.

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
