# Peer review of "A New Data-Preprocessing-Related Taxonomy of Sensors for IoT Applications"

_information, doi:10.3390/info13050241_

Round 1

Reviewer 1 Report

The authors proposed a new sensor taxonomy and compare the data filtering cost in the IOT applications. Moreover, some experiments are provided to measure the performance.

1. The discussion about sensor taxonomy should be improved, but the state-of-the-art results are not involved. 
2. Moreover, the readers would like to know the improvement from the result. Therefore, it is necessary to briefly describe that what is improved in this manuscript in abstract, and details should be stated in introduction and other sections.
3. The authors did some experiments to discuss the performance. Is it possible that implementing the proposed idea in the real world case? If yes, what is the major benefit? If not, what is the major difficult?

Author Response

We are grateful for the provided comments and believe that the manuscript has been greatly improved based on your feedback. We look forward to hearing whether you find this revision suitable for publication. We have revised the manuscript based on your corrections and suggestions. Please find next our replies, as well as the changes highlighted (in-blue text) over the manuscript's new version

Reviewer 2 Report

The following should be noted and corrected accordingly:

1. How practicable is your proposed model in real-time? 

2. Is it cost-efficient?

3. Some diagrams and terms are not properly explained.

4. Grammar is not up to standard and requires extensive re-editing

5. Are the formulas and numbers here generic or generated by you?

6. More keywords should be included

7. Related Works mentioned are too few compared to how broad the field of discourse is.

Study and consider the following related papers to embellish your paper:

• https://doi.org/10.1016/j.egyr.2022.02.304

• doi: 10.1109/TII.2021.3140109. 

• https://doi.org/10.1016/j.comcom.2021.09.029

Revisions are required.

Author Response

(The authors gave the same response as above.)

Round 2

Reviewer 1 Report

The authors addressed all concerns in the first submission. The revision looks good, and it could be accepted for publication.